# Phenotypic covariance across the entire spectrum of relatedness for 86 billion pairs of individuals

Kathryn E. Kemper [1 ✉], Loic Yengo [1], Zhili Zheng[1], Abdel Abdellaoui[2], Matthew C. Keller[3,4], Michael E. Goddard[5,6], Naomi R. Wray [1,7], Jian Yang [1] & Peter M. Visscher [1,7 ✉]

Attributing the similarity between individuals to genetic and non-genetic factors is central to genetic analyses. In this paper we use the genomic relationship ($\pi$) among 417,060 individuals to investigate the phenotypic covariance between pairs of individuals for 32 traits across the spectrum of relatedness, from unrelated pairs through to identical twins. We find linear relationships between phenotypic covariance and $\pi$ that agree with the SNP-based heritability ($\hat{h}^2_{SNP}$) in unrelated pairs ($\pi < 0.02$), and with pedigree-estimated heritability in close relatives ($\pi > 0.05$). The covariance increases faster than $\pi\hat{h}^2_{SNP}$ in distant relatives ($0.02 > \pi > 0.05$), and we attribute this to imperfect linkage disequilibrium between causal variants and the common variants used to construct $\pi$. We also examine the effect of assortative mating on heritability estimates from different experimental designs. We find that full-sib identity-by-descent regression estimates for height (0.66 s.e. 0.07) are consistent with estimates from close relatives (0.82 s.e. 0.04) after accounting for the effect of assortative mating.

[1] Institute for Molecular Bioscience, University of Queensland, Brisbane, QLD, Australia. [2] Department of Psychiatry, Amsterdam UMC, University of Amsterdam, Amsterdam, the Netherlands. [3] Department of Psychology & Neuroscience, University of Colorado at Boulder, Boulder, CO, USA. [4] Institute for Behavioral Genetics, University of Colorado at Boulder, Boulder, CO, USA. [5] Faculty of Veterinary and Agricultural Science, University of Melbourne, Parkville, VIC, Australia. [6] Biosciences Research Division, Department of Economic Development, Jobs, Transport and Resources, Bundoora, VIC, Australia. [7] Queensland Brain Institute, University of Queensland, Brisbane, QLD, Australia. ✉email: k.kemper@uq.edu.au; peter.visscher@uq.edu.au

The narrow-sense heritability ($h^2$) of a trait is the proportion of phenotypic variance that can be attributed to additive genetic effects[1]. It is a key population-dependent parameter in quantitative genetics and determines our ability to predict disease risk in medicine or the response to selection in agriculture. In practice $h^2$ is unknown but we can estimate it ($\hat{h}^2$) under various experimental designs and by invoking a range of assumptions, some of which are obvious (e.g. absence of non-additive genetic variance) and others more subtle (e.g. random mating and absence of genotype-environment covariance)[2]. Comparing heritability estimates across different experimental designs can cause confusion because the estimators can have different expectations, even when genetic effects are the only source of trait similarity between individuals. In Fig. 1 and Table 1, we give an overview of our study design and define the estimators used in this study.

Most simply, heritability can be estimated by comparing the observed resemblance between relatives to their expectations for a given genetic relationship[2]. Contrasting different relationship types then allows the separation of genetic and non-genetic components. In complex pedigrees, for example, the slope from the regression of the pairwise phenotypic covariance on genetic relationship provides an estimate of the genetic variance of a trait[3]. When variation due to known covariates are excluded from the phenotype[1], and phenotypes are standardised to unit variance, this slope is also an estimate of the trait heritability. Visscher et al.[4], for example, use this approach to estimate the heritability for height as 0.75. The approach is simple, but problems can arise when experiments violate assumptions underlying the estimation procedure. For example, Visscher et al.[4] find the intercept of their regression to be greater than zero, implying a phenotypic correlation among unrelated individuals. They attribute this observation to non-random (or assortative) mating[4]. Simple models can also be biased by non-additive genetic or common environmental effects, or other confounders which increase the phenotypic covariance among relatives beyond that expected.

In the absence of pedigree information, single nucleotide polymorphisms (SNPs) can be used to estimate genomic relationships between individuals ($\pi$). There are approximately $\frac{1}{2}N^2$ pairwise relationships between $N$ individuals within a population sample and, depending on the sample, most of these relationships are likely to be between (conventionally) unrelated individuals ($\pi < 0.02$). Unrelated individuals can be used to estimate the SNP-based heritability ($\hat{h}^2_{SNP}$) or the additive genetic effects captured by common SNPs. An advantage of this approach is that the SNP-based heritability has few biases from common environment and non-additive genetic effects[5], but a disadvantage is that $\hat{h}^2_{SNP}$ captures only part of the total genetic covariance due to imperfect tagging of causal variants by common SNPs[6]. Yang et al.[6] used a mixed linear model and restricted maximum likelihood (REML) to estimate $\hat{h}^2_{SNP}$ for height as 0.56 (s.e. 0.02). An equivalent estimate can be obtained using regression[7] and this implies that the slope from the regression of phenotypic correlation on genomic relationship ($\pi$) is about 0.56 for height, or about one- to two-thirds of the slope in close relatives[5,8]. Several studies have replicated these disparate heritability estimates in a single dataset from close and unrelated pairs using mixed linear models[9,10], but most have dichotomised genomic relationships by fitting a two component linear mixed model. In this paper, we sought to investigate how phenotypic correlation (and its regression slope) changes as a function of genomic relatedness within a population, across all pairs of individuals from nominally unrelated pairs through to monozygotic twins. In addition, we compare our heritability estimates with two other experimental designs, namely full-sib identity-by-descent (IBD) regression[11] and classic twin pair estimates.

Full-sib IBD regression[11] and the analysis of twin pairs are two alternative approaches to estimate $h^2$. Full-sib IBD regression (also referred to as full-sib regression[12]) has the advantage of avoiding confounding between genetic and environmental covariance in relatives by estimating within-family genetic variation (i.e. segregation variance or the genetic variation which arises from the random assortment of alleles at meiosis). The analysis exploits the fact that although full-sibs share on average 50% of their genome IBD, there is random segregation variation around this average, so that some pairs only share 40% of their genome IBD whereas other pairs share 60% of their genome IBD. A constraint of this type of analysis is that many tens of thousands of full-sib pairs are required for precise estimates due to the relatively small variance in IBD estimates. The second alternative design, a classic twin study, was once the mainstay of genetic analysis in humans and has been widely applied to many traits[13,14]. Classic twin analyses contrast phenotypic correlations between monozygotic ($r_{MZ}$) and dizygotic ($r_{DZ}$) twin pairs to estimate the heritability as $2(r_{MZ} - r_{DZ})$. A drawback of this approach is its heavy reliance on assumptions which are easily violated in close relatives, such as negligible non-additive genetic effects.

Random mating is an assumption underlying most approaches to estimate heritability[2] and is often violated in human populations. Assortative mating (AM) is a form of non-random mating that occurs when individuals with similar phenotypes tend to mate more often than expected by chance. In humans, it is reported for a number of traits, including height, body mass index (BMI), educational attainment (EA) and a range of psychiatric disorders[15,16]. The presence of AM is important for genetic analysis as AM causes causal variants across the genome to be correlated with one another (called gametic phase disequilibrium)[2]; increasing the genetic and phenotypic variance in a population as well as the changing the expected genetic and phenotypic covariance between different types of relative pairs[2]. Failure to account for AM can bias parameter estimates, lead to confusion about how much variation has been captured by identified variants, how well common disease can be predicted from polygenic risk scores and how much of the phenotypic similarity between relatives is due to environmental similarities.

In this study we examine the phenotypic covariance between pairs of individuals across the entire spectrum of relatedness, from (nominally) unrelated pairs, to distant relatives, full sibs and

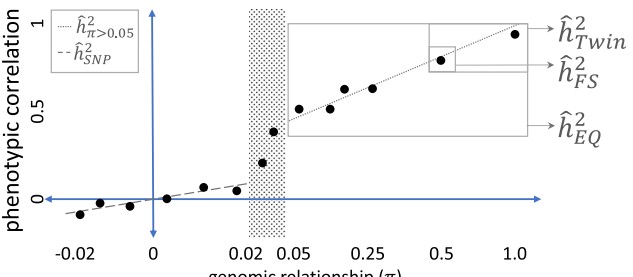

**Fig. 1 Schematic of the study design.** We use data from the UK Biobank to group pairs of individuals based on their genomic relationship ($\pi$), and calculate phenotypic correlations within 54 genomic relationship bins for 32 traits across the spectrum of relatedness. Subsets of the data are then used to estimate $\hat{h}$ under different designs, where the design-dependent estimates are defined in Table 1. From previous studies, we expect the slope of the regression of phenotypic correlation on genomic relationship in unrelated individuals ($\pi < 0.02$) to be less than that observed in close relatives.

**Table 1 Definitions of heritability and their use in this study.**

Population parameters

$h^2$ — Heritability; proportion of phenotypic variance attributable to additive genetic effects.

$h^2_{RM}$ — Heritability under random mating conditions, which is the usual definition of $h^2$ but is used here when contrasting it to $h^2$ under assortative mating (AM).

$h^2_{EQ}$ — Heritability after many generations of AM, i.e. after reaching equilibrium conditions, where $r$ is the correlation between mates under AM conditions.

Design-dependent estimators

$h^2_{SNP}$ — SNP-based heritability; proportion of phenotypic variance attributable to additive genetic effects associated with common SNPs. Estimated from linear regression in nominally unrelated individuals ($\pi < 0.02$, where $\pi$ is the genomic relationship).

$h^2_{\pi > 0.05}$ — Heritability estimated from linear regression in close relatives, where close relatives are defined by genomic relationships $> 0.05$.

$h^2_{FS}$ — Heritability estimated from full-sib identity-by-descent (IBD) regression, or the regression of phenotypic covariance on the realised relationship between full-sib pairs.

$h^2_{Twin}$ — Heritability estimated from a classic twin design by contrasting the phenotypic correlations between monozygotic and dizygotic twin pairs.

monozygotic twins. We use data from the UK Biobank and, due to the absence of pedigree information, quantify the genomic relationship ($\pi$) between ~ 86 billion pairs of individuals using 1.1 M HapMap3 SNPs. Our regressions of phenotypic correlation on genomic relationship show a slope equivalent to $\hat{h}^2_{SNP}$ in unrelated individuals ($\pi < 0.02$), and a slope equivalent to the pedigree-estimate heritability in close relatives ($\pi > 0.05$). We find that the increase in phenotypic correlation in distant relatives beyond that expected by $\pi\hat{h}^2_{SNP}$ can be reproduced by simulating incomplete linkage disequilibrium between causal variants and the variants used to construct $\pi$. We also untangle the influence of assortative mating on heritability estimates under a number of experimental designs, including complex pedigrees, a meta-analysis of IBD regression from ~100 K full-sib pairs and published twin correlations.

## Results

**Phenotypic covariance as a function of genomic relationship.** We report the phenotypic covariance between pairs of individuals of British and Western European ancestry in the UK Biobank for 32 quantitative or ordered categorical traits, with up to 416 K observations per trait (Supplementary Table 1). We created a genomic relationship matrix among the 417 K individuals with 1 or more phenotypes using ~1.1 million HapMap3 SNPs to obtain over 86 billion pairwise relationships [i.e. ~½(417 K)²]. Genomic relationships ($\pi$) were divided into 54 relationship bins based on the observed distribution of $\pi$, where there were between 7.6 billion and 159 pairs per bin (Supplementary Fig. 1). Phenotypes were pre-adjusted for known covariates including age and sex, as part of the quality control process of the phenotype data. We additionally fitted models to the phenotypes that aimed to account for technical, genetic and geographic stratification (Supplementary Table 2). Accounting for technical stratification included fitting genotyping batch. Genetic stratification was accounted for by fitting 25 principal components (PCs) from the genomic relationship matrix. We recently showed that complex traits show geographic clustering in the UK Biobank sample[17] and hence we account for geographic stratification by fitting birth contemporary group (CG) as a factor based on 378 local authority areas. Traits varied considerably in the variation attributable to technical, demographic and genetic factors, and there was some confounding between these effects. Hence, we assessed the effect of CG after accounting for PCs and all other covariates (see model (4), 'Methods'). We find, for example, both height and educational attainment (EA) displayed modest stratification ($R^2 > 0.5\%$) for both CG and PCs (Supplementary Figure 2).

The estimated phenotypic correlation for a pair of individuals is the product of their standardised phenotypes. We calculated the average correlation for all pairs within each of the 54

relationship bins (see 'Methods'). We investigated the effect of geographic and genetic stratification on phenotypic covariance using 4 models for pre-correction of the phenotype (Supplementary Fig. 3). Fitting more factors (either CG or PCs) generally decreased the phenotypic correlation between pairs, where the variance explained ($R^2$) per factor determine the magnitude of reduction in covariance. For example, traits with little influence from either CG or PCs, such as bone mineral density, showed little change in phenotypic correlation between the 4 models fitted to the data. In contrast, the correlations reduced for EA and height when fitting PCs only, CG only or fitting both PCs and CG. We took a conservative approach and focus our results on the model accounting for both genetic and geographic stratification, irrespective of the variance these covariates explained.

The phenotypic correlation across the entire relatedness spectrum is shown for height, EA and body mass index (BMI) in Fig. 2. Overall, the relationship is not linear, which is expected because previous studies indicated that the variance captured by common SNPs ($\hat{h}^2_{SNP}$) is approximately one-third to two-thirds of the variance explained in pedigree (or close relatives)[5]. Our analysis identifies three trends: (i) a steady linear relationship between phenotypic correlation and genomic relationship for unrelated individuals ($\pi < 0.02$), (ii) an accelerated rate of increase in phenotypic correlation for distant relatives ($0.02 < \pi < 0.05$), and (iii) a second approximately linear increase in correlation among close relatives ($\pi > 0.05$). The heritability obtained from the regression of phenotypic correlation on genomic relationship using the relationship bins is equivalent to individual-level Haseman-Elston (HE) regression[7] for the sections of the distribution that are linear. We use this property, combined with simulations, to investigate each of the three trends observed in the phenotypic correlation function.

**The regression of phenotypic correlation on genomic relationship in unrelated individuals estimates the SNP-based heritability ($\hat{h}^2_{SNP}$).** The slope for the regression of phenotypic correlation on genomic relationship in unrelated individuals estimates genetic variation captured by common SNPs.[7] We fitted a weighted linear model in R[18] for the unrelated bins ($\pi < 0.02$) to show that, for the majority of traits, the slope of the regression line is consistent with estimates of $\hat{h}^2_{SNP}$ from HE regression based on a subset of individual-level data[7] (Supplementary Table 3; Supplementary Fig. 4). We also explored the effect of fitting both CG and PCs on $\hat{h}^2_{SNP}$ and found for traits where these factors explained a relatively large proportion of the trait variance ($R^2 > 0.5\%$) there was a significant reduction in $\hat{h}^2_{SNP}$ when CG and PCs were fitted (Supplementary Fig. 5). For example, $\hat{h}^2_{SNP}$ decreases from 0.63 (s.e. 0.005) to 0.54 (s.e. 0.004) when both PCs

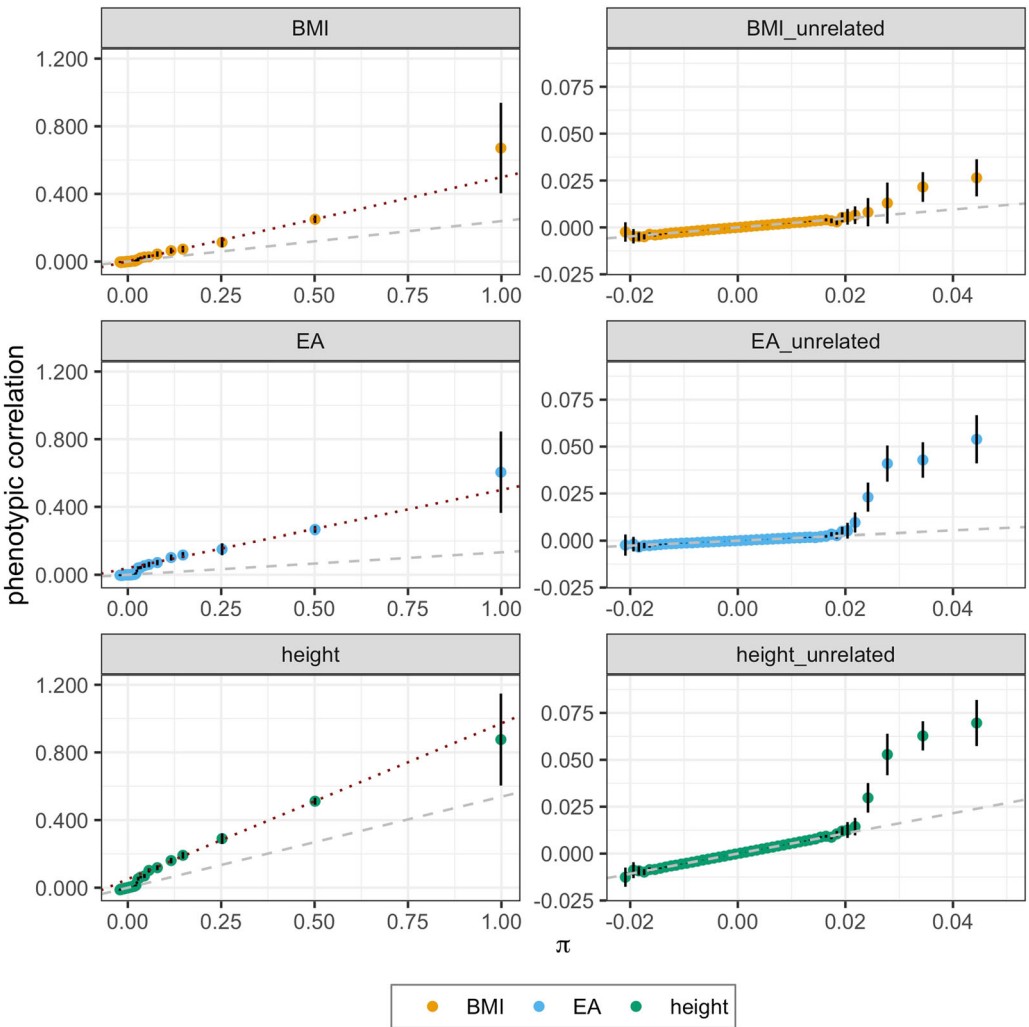

**Fig. 2 Phenotypic correlation between pairs as a function of genomic relationship ($\pi$).** Traits shown are body mass index (BMI), educational attainment (EA) and height; with either all pairs (left) or highlighting the regression of phenotypic covariance on genomic relationship in un- and distantly related ($\pi < 0.05$) pairs (right). Points indicate the mean correlation from all data, with vertical bars showing 95% confidence intervals ($\pm 1.96$ s.e., where standard errors are derived from a blocked jacknife procedure with 100 blocks of individuals). Dashed grey line shows the best fit line for unrelated individuals (interpreted as the SNP-heritability, $\hat{h}^2_{SNP}$) and the red dotted line shows the fit line for close relatives (interpreted as $\hat{h}^2_{\pi>0.05}$).

and CG are fitted for height. This suggests that PCs and CG can remove proportionately more genetic variance than phenotypic variance in some instances.

**Incomplete linkage disequilibrium recapitulates the increase in phenotypic correlation in distant relatives ($0.02 < \pi < 0.05$).** We observed that the phenotypic correlation increases more rapidly than that predicted by the SNP-based heritability in distant relatives ($0.02 < \pi < 0.05$; Fig. 2) and reasoned that increased linkage disequilibrium (LD) for close relatives between rare causal variants and the common SNPs used to estimate $\pi$ could underlie this trend. Under a simple AE model (i.e. additive genetic effects plus random environmental deviations), we used simulation to investigate this possibility and confirmed that incomplete LD could reproduce the observed increase in phenotypic correlation in distant relatives (Supplementary Note 1). In real data, other factors such as common environmental effects or non-additive genetic effects may also influence the increase, although the contribution of non-additive genetic variance to the phenotypic covariance in distant relatives is negligible because non-additive genetic variance is a function of $\pi^2$ (Supplementary Note 2). Thus simulations suggest that LD (in the absence of other effects) can

cause an increase in phenotypic covariance larger than $\pi\hat{h}^2_{\pi>0.05}$ for distant relatives ($0.02 < \pi < 0.05$). Genetic architectures where a substantial proportion of additive genetic variance is due to imperfectly tagged rare variants provide a parsimonious explanation of the observations.

**Heritability ($\hat{h}^2_{\pi>0.05}$) in close relatives.** Our bin-based analysis can be described as a weighted HE regression. We compared this approach with an individual-level HE regression of phenotypic correlation on genomic relationship from close relatives ($\pi > 0.05$). Individual-level HE regression estimated two variance components which were equivalent to the intercept and slope of the bin-based weighted HE-regression (see 'Methods').

We find our bin-based weighted HE regression to be a good approximation of the heritability estimated from individual-level HE regression for most traits (Supplementary Fig. 6; Supplementary Table 4). We tested if the intercept estimated in our weighted HE regression analysis was different from zero and found significant evidence ($p < 0.05/32$) for height, income, lung capacity (FVC, forced vital capacity) and EA. Note that a non-zero intercept could be caused by a range of factors, including common environmental effects or AM[4]. We chose to investigate

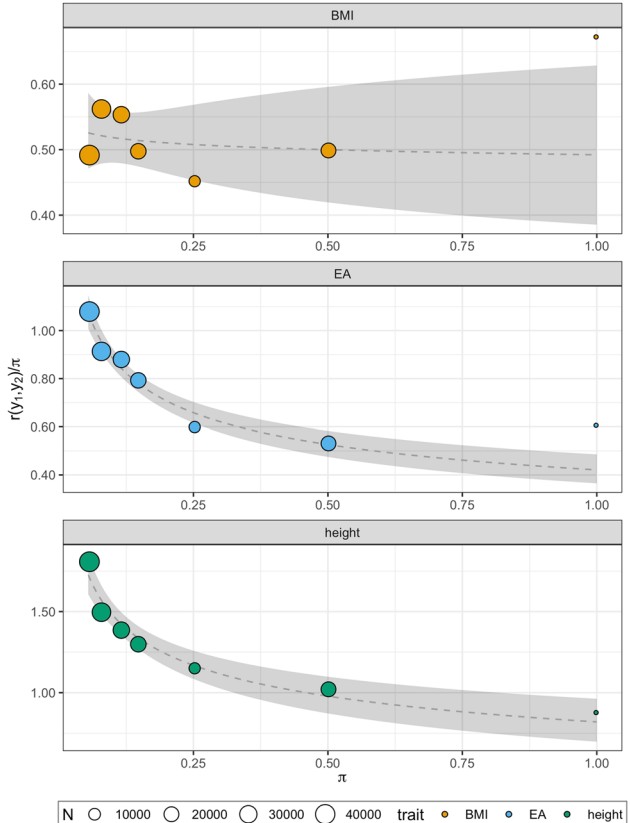

**Fig. 3 Modelling of assortative mating in close relatives ($\pi > 0.05$).** Traits shown are body mass index (BMI), educational attainment (EA) and height. Points show the phenotypic correlation for close relatives divided by the mean genomic relationship ($\pi$) and $\pi$, where $N$ = number of pairs contributing to each point. The fitted function is shown with 95% confidence intervals ($\pm 1.96$ s.e.) in grey.

the influence of AM on our heritability estimates for close relative pairs and then use full-sib IBD regression to consider the influence of common environmental effects.

**Accounting for assortative mating in relatives.** AM is expected to increase the phenotypic correlation for relative pairs, where the effect depends on the pedigree relationship between the pairs of relatives, the magnitude of the phenotypic correlation between mates ($r$) and the equilibrium heritability of the trait ($h^2_{EQ}$)[19]. In the absence of common environmental effects and non-additive genetic variation, the relationship between the phenotypic correlation of relatives and their genetic relatedness is known[20]. We used the realised genomic relationship (i.e., $\pi$) as a proxy for pedigree relationships and the expectation for the correlation between relatives under AM to estimate the heritability after many generations of assortative mating (equilibrium heritability, $\hat{h}^2_{EQ}$) and infer $\hat{r}$ without directly observing spouse pairs. Note that our $\hat{r}$ is estimated on the basis of primary phenotypic assortment and is not influenced by factors which may influence $\hat{r}$ when it is calculated from spousal pairs, e.g. convergence of phenotypes or social homogamy.

We applied our model to a subset of 14 heritable traits ($\hat{h}^2_{SNP} > 0.1$) with a large number of records ($N > 400$K) and find negligible AM in most cases (Supplementary Table 5). The exceptions were height and EA where estimates of spousal correlation were significantly different from zero (height, $\hat{r} = 0.24$ s.e. 0.04; EA $\hat{r} = 0.60$ s.e. 0.19), and consistent with other

studies using genomic information to infer $r$ (e.g. height, $\hat{r} = 0.200$ s.e. 0.004; EA, $\hat{r} = 0.654$ s.e. 0.014)[21]. The equilibrium estimates of heritability for height (0.82, s.e. 0.04) and EA (0.42, s.e. 0.04; Supplementary Table 5) were lower than that from the weighted HE regression in close relatives which did not account for AM ($\hat{h}^2_{\pi>0.05}$; Supplementary Table 4). Figure 3 shows the observed curvilinear relationship between phenotypic covariance (scaled by $\pi$) and $\pi$ for height and EA, and the contrasting (almost linear) relationship for BMI. Consistent with negligible AM for BMI, we find no difference for the heritability estimated using either the linear regression or the model accounting for AM ($\hat{h}^2_{\pi>0.05} = 0.50$, s.e. 0.03; $\hat{h}^2_{EQ} = 0.49$, s.e. 0.07; Supplementary Tables 5 and 6). Attempting to account for AM increased standard errors of heritability estimates for most traits when the spousal correlations were not significantly different from zero. Hence, we report heritability from close relatives as $\hat{h}^2_{\pi>0.05}$ for most traits, and as $\hat{h}^2_{EQ}$ for height and EA where significant spousal correlations were detected (Table 2).

**Full-sib IBD regression estimates heritability free from environmental covariance.** The phenotypic correlation between pairs of full-sibs ($r_{FS}$) can be due to both genetic and environmental factors, where full-sib IBD regression partitions this correlation into genetic ($h^2_{FS}$) and environmental ($c^2_{FS}$) components. Hence $r_{FS} = c^2_{FS} + \overline{IBD}h^2_{FS}$ (where $\overline{IBD} = 0.5$, i.e. the average coefficient of genomic relationship for full-sibs). Models can be specified at the individual[22] or sib-pair level[12], and we show that these two approaches are equivalent (Supplementary Note 3). The advantage of parameter estimates from full-sib IBD regression analysis is that $h^2_{FS}$ unaffected by environmental covariance within sibling pairs[11], and this analysis is independent of other estimates ($\hat{h}^2_{\pi>0.05}$ or $\hat{h}^2_{EQ}$) as it uses within family information.

An individual-level model and REML was used to estimate $\hat{c}^2_{FS}$ and $\hat{h}^2_{FS}$ for almost 20 K full-sib pairs in the UK Biobank (Table 2). In accordance with expectations[11], standard errors were large for all traits and thus we improved the precision of our estimates by meta-analysis of our results with others available in the literature for height, BMI and EA[12,22] (Table 3). Although the meta-analysis includes ~100 K pairs for height and BMI, and ~50 K pairs for EA; standard errors remained in the order of 0.10 and 0.05 for $\hat{h}^2_{FS}$ and $\hat{c}^2_{FS}$, respectively, thus limiting our power. Despite this limitation, we found the full-sib IBD heritability estimates for height to be significantly lower than our estimate of $\hat{h}^2_{EQ}$ from close relatives ($\hat{h}^2_{FS} = 0.66$ s.e. 0.07; $\chi^2_1 = 4.7$, $p = 0.03$). We also find the environmental covariance term to be significantly greater than zero for both height ($\chi^2_1 = 16$, $p = 6.3 \times 10^{-5}$) and EA ($\chi^2_1 = 4.7$, $p = 0.03$).

**Adjusting parameter estimates for assortative mating.** The analysis of close relatives detected evidence consistent with the effects of AM for height and EA in the UK Biobank. We, therefore, decided to explore the influence of AM on the interpretation of the full-sib IBD regression results, and also the heritability estimated under another widely used experimental design, namely a classical twin study of monozygotic and dizygotic twin pairs. We show with theory and simulation that the expectation from full-sib IBD regression of $h^2_{FS}$ and $c^2_{FS}$ under AM are $h^2_{EQ}(1 - rh^2_{EQ})$ and $h^2_{EQ} - h^2_{FS}$, respectively (Supplementary Note 3). Thus, $h^2_{FS}$ is an estimate of neither the equilibrium ($h^2_{EQ}$) nor the random mating ($h^2_{RM}$) heritability but the random mating genetic variance scaled by the phenotypic variance in the current population. In addition, $c^2_{FS}$ captures the gametic phase

**Table 2 Traits with moderate-high heritability ($\hat{h}^2_{SNP} > 0.1$) and total number of records >400 K with estimates for the variance captured by common SNPs ($\hat{h}^2_{SNP}$ with standard error s.e.) from unrelated pairs ($\pi < 0.02$; where $\pi$ is the genomic relationship), the heritability from close relatives ($\pi > 0.05$; either $\hat{h}^2_{\pi > 0.05}$ or $\hat{h}^2_{EQ}$), the observed full-sib phenotypic correlation ($\hat{r}_{FS}$), and its partitioning into common environment ($\hat{c}^2_{FS}$) and additive genetic effects (heritability, $\hat{h}^2_{FS}$).**

| Trait | N.records | Unrelated pairs ($\pi$<0.02) | | | Close relative pairs ($\pi$>0.05) | | | Full-sib pairs ($\tau_{IBD} \sim 0.50$) | | | | | |
|---|---|---|---|---|---|---|---|---|---|---|---|---|---|
| | | N.pair (×10⁹) | $h^2_{SNP}$ | s.e. | N.pair | $h^2_{\pi>0.05}$ ($h^2_{EQ}$)ᵃ | s.e. | N.pair | $\hat{r}_{FS}$ | $\hat{c}^2_{FS}$ | s.e. | $\hat{h}^2_{FS}$ | s.e. |
| height | 416,131 | 86.6 | 0.538 | 0.004 | 156,335 | 0.819 | 0.035 | 19,954 | 0.528 | 0.231 | 0.059 | 0.597 | 0.116 |
| metRate | 409,465 | 83.8 | 0.306 | 0.003 | 151,279 | 0.623 | 0.027 | 19,368 | 0.329 | 0.034 | 0.080 | 0.596 | 0.160 |
| pcc | 404,144 | 81.7 | 0.299 | 0.003 | 147,399 | 0.541 | 0.031 | 18,821 | 0.288 | 0.299 | 0.084 | -0.023 | 0.168 |
| wbcc | 404,211 | 81.7 | 0.266 | 0.005 | 147,386 | 0.297 | 0.021 | 18,816 | 0.205 | 0.134 | 0.089 | 0.140 | 0.178 |
| sit:stand | 414,868 | 86.0 | 0.254 | 0.003 | 155,416 | 0.430 | 0.028 | 19,841 | 0.244 | 0.156 | 0.086 | 0.177 | 0.173 |
| rbcc | 404,317 | 81.7 | 0.249 | 0.003 | 147,527 | 0.490 | 0.026 | 18,841 | 0.246 | 0.172 | 0.087 | 0.148 | 0.174 |
| BMI | 415,308 | 86.2 | 0.239 | 0.003 | 155,692 | 0.497 | 0.026 | 19,885 | 0.270 | -0.132 | 0.084 | 0.807 | 0.167 |
| ecc | 402,539 | 81.0 | 0.231 | 0.004 | 146,276 | 0.255 | 0.022 | 18,682 | 0.175 | -0.096 | 0.092 | 0.543 | 0.184 |
| bodyFat | 409,550 | 83.9 | 0.223 | 0.002 | 151,376 | 0.518 | 0.026 | 19,374 | 0.273 | 0.067 | 0.085 | 0.413 | 0.169 |
| skinTan | 408,456 | 83.4 | 0.199 | 0.002 | 150,381 | 0.392 | 0.029 | 19,188 | 0.209 | 0.059 | 0.089 | 0.295 | 0.178 |
| skinColour | 411,315 | 84.6 | 0.187 | 0.002 | 152,413 | 0.358 | 0.027 | 19,480 | 0.192 | -0.066 | 0.088 | 0.518 | 0.176 |
| W:H | 416,167 | 86.6 | 0.165 | 0.002 | 156,364 | 0.370 | 0.023 | 19,958 | 0.197 | -0.096 | 0.088 | 0.588 | 0.175 |
| handGrip | 416,124 | 86.6 | 0.144 | 0.002 | 156,279 | 0.350 | 0.027 | 19,957 | 0.182 | 0.114 | 0.089 | 0.137 | 0.178 |
| EA | 413,496 | 85.5 | 0.133 | 0.002 | 153,925 | 0.421 | 0.043 | 19,736 | 0.295 | 0.222 | 0.083 | 0.144 | 0.165 |

The number of pairs (N.pair) contributing to each estimate is shown.
Traits include height, metabolic rate (metRate), platelet cell count (pcc), white blood cell count (wbcc), ratio of sitting to standing height (sit:stand), red blood cell count (rbcc), body mass index (BMI), eosinophil cell count (ecc), body fat percentage (bodyFat), skin tanning (skinTan), skin colour (skinColour), waist-to-hip ratio (W:H), hand grip strength (handGrip) and educational attainment (EA). Further trait details are given in Supplementary Table 1. For traits with significant assortative mating (height and EA), the value is the equilibrium heritability ($\hat{h}^2_{EQ}$).
ᵃ$\hat{h}^2_{\pi>0.05}$ presented for most traits.

disequilibrium variance generated by AM plus the environmental covariance between pairs. Similarly, we show that the expectation for the heritability estimated under a twin design is $h^2_{EQ}(1 - r h^2_{EQ})$ and that AM also inflates the environmental covariance from this design (Supplementary Note 3). Thus, in the presence of AM the heritability estimated using different experimental designs are not directly comparable to $h^2_{EQ}$, and the environmental covariance is inflated by the genetic variance generated by AM.

The equilibrium heritability ($\hat{h}^2_{EQ}$) reflects the genetic variation in the current population. It can be estimated from twin and full-sib designs if the correlation between mates ($r$) is known or precisely estimated. We obtained estimates of $r$ using published phenotypic correlations between spousal pairs and adjusted the estimates from twin and full-sib designs to estimate $h^2_{EQ}$. Namely we adjusted the (i) full-sib IBD regression meta-analysis results, and (ii) classic twin estimates from published meta-analysis[23,24] for AM (Supplementary Note 4). Each adjusted estimate is compared to our estimate of the equilibrium heritability ($\hat{h}^2_{EQ}$) from close relatives for height and EA (Table 4). In contrast to $\hat{h}^2_{FS}$, there is no significant difference between $\hat{h}^2_{EQ}$ and the adjusted full-sib IBD regression estimate for height ($\chi^2_1 < 0.01$, $p > 0.05$). We also found that the adjusted twin estimate to be significantly larger than $\hat{h}^2_{EQ}$ ($\chi^2_1 = 5.9$, $p = 0.01$). For EA the standard errors for adjusted estimates were large and confidence intervals spanned much of the parameter space. Unfortunately this prevents meaningful comparisons for this trait (Table 4).

## Discussion

We present the phenotypic covariance across the full spectrum of genomic relatedness for 32 quantitative and ordered categorical traits. Our approach is simple and unencumbered by assumptions about the underlying genetic constitution of a trait. We observe that phenotypic correlation increases with genomic relationship from nominally unrelated pairs through to monozygotic twins. The relationship is composed of two linear sections, one in unrelated ($\pi < 0.02$) and one in close relative ($\pi > 0.05$) pairs, with a joining section of sharply increasing phenotypic covariance in distant relatives ($0.02 < \pi < 0.05$). We use simulation to argue that a simple AE model of genetic architecture, where A is the additive genetic effect and E is the unique environment effect, and incomplete linkage disequilibrium between causal variants and the SNPs used to construct the genomic relationship matrix can generate the observed pattern in phenotypic covariance (Supplementary Note 1). In accordance with expectations[5,8], common SNPs capture between one- and two-thirds of the heritability estimated from close relatives for a range of quantitative traits (Table 2). These results illuminate the continuous relationship between the phenotypic correlation and the genomic relationship, particularly for distant relatives not normally considered by pedigree studies ($0.02 < \pi < 0.08$, i.e., more distant than 1ˢᵗ cousins), and suggest that $\pi < 0.02$ is a good threshold for studies wishing to estimate $\hat{h}^2_{SNP}$ (i.e. the phenotypic variation tagged by common SNPs in the population).

We find that non-random mating had a measurable effect on the phenotypic correlation of close relatives for some traits. The phenotypic correlation of close relative pairs was higher than what would be expected under a simple AE model, and the AE model predicted that unrelated pairs to have a non-zero phenotypic correlation for height, income, lung capacity and EA. These results replicate previous observations for height using pedigree data[4]. We modelled these effects using the expected correlations for close relative pairs under equilibrium conditions for AM and inferred significant correlations between mates for height ($\hat{r} = 0.24$ s.e. 0.04) and EA ($\hat{r} = 0.60$ s.e. 0.19). These results are

**Table 3 Meta-analysis of full-sib IBD regression results for height, body mass index (BMI) and educational attainment (EA).**

| Trait | Study | N.pair | $\hat{r}_{FS}^a$ | s.e. | $\hat{c}_{FS}^2$ | s.e. | $\hat{h}_{FS}^2$ | s.e. |
|---|---|---|---|---|---|---|---|---|
| Height | This paper | 19,954 | 0.53 | 0.01 | 0.23 | 0.06 | 0.60 | 0.12 |
| | Young et al.[12] | 64,847 | 0.41 | 0.01 | 0.07 | 0.05 | 0.68 | 0.10 |
| | Hemani et al.[22] | 20,240 | 0.43 | 0.01 | 0.08 | 0.07 | 0.69 | 0.14 |
| | Meta-analysis | 105,041 | 0.44 | <0.01 | 0.12 | 0.03 | 0.66 | 0.07 |
| BMI | This paper | 19,885 | 0.27 | 0.01 | −0.13 | 0.08 | 0.81 | 0.17 |
| | Young et al.[12] | 56,461 | 0.27 | 0.01 | 0.08 | 0.06 | 0.39 | 0.12 |
| | Hemani et al.[22] | 20,240 | 0.31 | 0.01 | 0.10 | 0.08 | 0.42 | 0.17 |
| | Meta-analysis | 96,586 | 0.28 | <0.01 | 0.03 | 0.04 | 0.50 | 0.08 |
| EA | This paper | 19,736 | 0.29 | 0.01 | 0.22 | 0.08 | 0.14 | 0.16 |
| | Young et al.[12] | 32,542 | 0.36 | 0.01 | 0.16 | 0.07 | 0.40 | 0.15 |
| | Meta-analysis | 52,278 | 0.34 | <0.01 | 0.19 | 0.06 | 0.28 | 0.11 |

Shown is the number of sib-pairs (N.pair) in each study, inferred full-sib correlation ($\hat{r}_{FS}$), common environmental effect ($\hat{c}_{FS}^2$) and heritability ($\hat{h}_{FS}^2$) with standard errors (s.e.).
[a]in each study, we calculated the full-sib correlation from the given estimates of common environmental and genetic effects from full-sib regression as $\hat{c}_{FS}^2 + \frac{1}{2}\hat{h}_{FS}^2$. Standard errors were estimated using the approximation $\sqrt{(1 - \hat{r}_{FS}^2)/N}$, where $N$ is the number of pairs.

**Table 4 Estimates of the equilibrium heritability for height and EA from different experimental designs, using adjustments for assortative mating where appropriate[a].**

| Data source | Experimental design | Height (s.e) | EA (s.e) |
|---|---|---|---|
| This paper | Close relatives | 0.82 (0.04) | 0.42 (0.04) |
| Meta-analysis, this paper | Full sib IBD regression[a] | 0.81 (0.10) | 0.33 (0.12) |
| Meta-analysis, literature[23,24] | Classic twin pairs[a] | 0.93 (0.03) | 0.49 (0.08) |

[a]full details of parameter expectations under AM and adjustments are given Supplementary Note 4.

concordant with a second study inferring the correlation between mates using the regression coefficient of one individual's phenotype on their partner's genomic predictor for these traits (height $\hat{r} = 0.20$ s.e. 0.007; EA $\hat{r} = 0.65$ s.e. 0.014)[21], and several other studies estimating the phenotypic correlation between partners for height[4,25–27].

We detailed the real effects of AM on parameter estimates from close relatives, and highlight biases that may occur if these effects are not fully considered. AM changes both the genetic and phenotypic variance parameters and, depending on experimental design, the expected heritability estimate for the trait. For example, ignoring AM and fitting a linear regression of phenotypic correlation on genomic relationship for close relatives upwardly biased heritability estimates in our study (i.e. $\hat{h}_{\pi>0.05}^2$ was upwardly biased for height). We show that there is no simple expectation for the heritability estimated under complex pedigrees (Supplementary Note 4). In contrast, the expectation of the heritability obtained from full-sib IBD regression and classic twin studies is neither equilibrium nor random mating heritability, but rather the random mating genetic variance divided by the equilibrium phenotypic variance (Supplementary Notes 3 and 4). Thus, we cannot directly compare heritability estimates from different experimental designs. Also, the common environmental variance estimated under full-sib IBD regression and classic twin estimates can be inflated by genetic variance under AM. This may lead to an overstatement of the importance for common environmental effects in studies where AM is ignored. We advise caution when comparing past estimates of heritability when AM is ignored or improperly modelled for traits such as height and EA.

We estimate the equilibrium heritability for height from three types of experimental design by adjusting results from our paper and the literature for AM (Table 4). Our estimate of the equilibrium heritability from close relatives ($\hat{h}_{EQ}^2 = 0.82$ s.e. 0.04) is consistent with other studies modelling common environment and AM effects in close relatives[25,28]. For example, we meta-analysed estimates from Swedish full- and half-sibling raised together and apart to estimate $\hat{h}_{EQ}^2$ as 0.77 (s.e. 0.005)[28]. Our estimate of 0.82 could be inflated by other confounding factors (see discussion below) but the effects should be minimal unless the confounding effects are directly proportional to $\pi$. We observe that the twin estimate assessing the equilibrium heritability for height is significantly greater than our estimate of $\hat{h}_{EQ}^2$ from close relatives ($\chi_1^2 = 5.9$, $p = 0.01$; Table 4). This finding adds further to evidence for the systematic inflation of heritability estimates from classic twin studies[28–30]. In practice, this means monozygotic twins are more similar than expected under an ACE (additive genetic, common environment and random environment) model including the effects of AM[28].

Other potential sources of covariance between relatives, such as common environment, non-additive genetic effects and associative (or indirect) genetic effects are not directly considered in our model. Our estimates of $\hat{h}_{EQ}^2$ and $\hat{h}_{\pi>0.05}^2$ could therefore be upwardly biased by these factors. In particular, associative effects may be confounded with $\pi$ (the genomic relationship) and these effects may be important sources of covariance for traits such as EA[26,31].

Associative effects[32] are a covariance source that can lead to heritable components in the environment which might be confounded with $\pi$. Associative effects[32] occur when the phenotype of an individual is dependent on the phenotype of others. This can cause genotype-environment covariance; where the environmental effect of an individual is influenced by the phenotype of a relative, therefore creating a correlation between the individual's genotype and it's environment. Traits influenced by associative effects include aggression or competition amongst animals or plants[32], and possibly many behavioural traits in humans. Strong evidence has been reported for the presence of associative effects for EA[31,33]. Kong et al.[33], for example, attributed about 50% of the variance in EA explained by alleles transmitted from parents to offspring to associative effects (i.e. 'genetic nurture'). A second type of associative effects may also be important for the full-sib IBD regression results. These are sibling effects[34], where the phenotype of one sibling influences the second sibling. If present sibling effects have the potential to upwardly bias $\hat{h}_{FS}^2$ (Supplementary Note 5). Ideally, models to estimate genetic effects would

include associative effects. However, models including associative and additive genetic effects become highly parameterised and require specific experimental designs for their estimation (Supplementary Note 5). Lee et al.[31] found that the inflation of genome-wide association effects, compared to within family estimates, could be fully explained by AM for height but not for EA. This suggests that accurate modelling of EA requires accounting for both AM and associative effects.

Covariance between relative pairs could also increase due to non-additive genetic effects. Non-additive genetic variance results from interactions between alleles within (dominance) or between (epistasis) loci[35]. To explore the impact of additive-by-additive epistatic effects, we use simulation to show that the expected increase in phenotypic covariance with genomic relationship is proportional to $\pi^2$, i.e. a quadratic relationship (Supplementary Note 2). Subsequent tests of our subset of 14 phenotypes found no support for significant additive-by-additive genetic variance ($p > 0.05/14$). However, our study had sufficient power to detect only large additive-by-additive effects ($>0.45\sigma_P^2$, where $\sigma_P^2$ is the phenotypic variance) and we caution that non-linear effects are also susceptible to confounding with other non-linear factors affecting covariance in close relatives, such as shared environment or associative effects (Supplementary Note 2).

Young et al.[18] recently used family data from Iceland to obtain heritability estimates for 14 traits, including height (0.55 s.e. 0.04), BMI (0.29 s.e. 0.06) and EA (0.17 s.e. 0.09). These estimates are lower than many pedigree-based estimates[25,28], including those presented here for $\hat{h}_{\pi>0.05}^2$ (Table 2). Young et al.[18] use their RDR (relatedness disequilibrium regression) method, which is a generalisation of full-sib IBD regression, to estimate within-family genetic variance. Thus the RDR method potentially measures a quantity equivalent to $h_{EQ}^2\left(1 - rh_{EQ}^2\right)$, or the random mating genetic variance scaled by the phenotypic variance in the current population. Applying a correction to the RDR estimates (following the method outlined for full-sib IBD regression, Supplementary Note 4), we obtain estimates for the equilibrium heritability of 0.65 (0.06) for height and 0.18 (0.11) for EA. These estimates remain significantly lower than $\hat{h}_{EQ}^2$ for both height ($\chi_1^2 = 5.4$, $p = 0.02$) and EA ($\chi_1^2 = 3.9$, $p = 0.05$). Similarly, the RDR estimate for BMI is lower than our estimate using close relatives (0.50 s.e. 0.03) and other estimates which are free from environmental confounders (i.e. full-sibs raised apart[28]; 0.44 s.e. 0.04). Although Young et al.[18] conclude that common environmental effects have inflated past estimates, here we highlight that AM is also important for the interpretation of their results. The effects of AM may partially explain the differences between the results from the RDR method and other estimates.

Our paper makes use of identity-by-state (IBS) relationships to estimate heritability in close relatives. Ideally, these inferences would be made based on the proportion of the genome IBD (where IBD alleles are IBS and inherited from a common ancestor[36]) rather than IBS relationships[37]. IBD relationships are ideal because (true) IBD sharing is independent of the genotyped markers. However, there are challenges associated with the accurate estimation of the proportion IBD in relatives without pedigree information. Hill and White[38], for example, highlight that the proportion IBD estimated from the detection of shared segments can underestimate the true value when small segments are missed. The estimation and use of IBD and IBS relationships in close and distant relatives warrants further scrutiny.

We use an indirect assessment of AM based on the ability to detect inflation in phenotypic covariance in relatives to estimate significant AM for height ($\hat{r} = 0.24$ s.e. 0.04) and EA ($\hat{r} = 0.60$ s.e. 0.19). These results are consistent with a recent paper examining gametic phase disequilibrium (i.e. correlations

between trait-increasing alleles at distant loci) that found evidence consistent with AM for height and EA[39]. Robinson et al.[21] also infer a correlation between partners for height (0.20, s.e. 0.007) and EA (0.65, s.e. 0.014) under some assumptions and using the regression of genetic predictors on partner's phenotype. Similar to Robinson et al., we also find that the inferred correlation between mates for EA using genetic information is higher than Robinson's reported phenotypic correlation between (likely) partners in the UK Biobank (0.41 s.e. 0.01)[21]. Further, our meta-analysis of the phenotypic correlation between known partners in the literature (0.42 s.e. 0.01) supports a phenotypic correlation between spouses of around 0.4. Robinson et al.[21] suggests that the differences between the inferred and observed correlations could arise if there was indirect assortment on EA, that is direct assortment occurs on a trait genetically correlated with EA (such as intelligence). It may also indicate that equilibrium conditions have not been reached for EA, or that associative effects are influencing estimates of genetic variance. Robinson et al.[21] also find evidence of assortment for BMI (0.14 s.e. 0.007) and other metabolic traits which was not replicated in this study, nor in Yengo et al.[39]. However, the expected inflation in additive genetic variance and heritability for BMI caused by the magnitude of AM reported by Robinson et al.[21] is about 7 and 4%, respectively. Standard errors on our estimates of these parameters are of a similar magnitude as these effects and suggest insufficient power to detect the weak assortment for BMI reported by Robinson (Supplementary Table 5).

In summary, examining the change in phenotypic correlation between pairs of individuals as a function of their genomic relationship provides a simple approach to estimate the proportion of phenotypic covariance due to additive genetic effects. We used this approach to observe the change in phenotypic covariance across the entire spectrum of genomic relationships, from (nominally) unrelated pairs through to monozygotic twins. We observed two approximately linear sections to the distribution, one predicted by $\hat{h}_{SNP}^2$ in unrelated individuals ($\pi < 0.02$) and one predicted by $\hat{h}_{\pi>0.05}^2$ close in relatives. We used simulation to show that the correlation in distant relatives can be recapitulated by incomplete linkage disequilibrium between causal variants and the SNPs used to calculate genomic relationships. Finally, we detail the influence of AM on heritability estimates in close relatives and advise caution when directly comparing between different experimental designs for traits showing AM, such as height. Our results show that common environment effects estimated in several experimental designs are completely confounded with the genetic variance generated by assortative mating. We suggest that this may have led to an overemphasis on the importance of shared environment for traits undergoing AM.

## Methods

**Ethical compliance.** The North West Centre for Research Ethics Committee granted ethics approval the UK Biobank study for (11/NW/0382). Participants in the study provided signed electronic consent upon recruitment. This research is approved under the University of Queensland human ethics committee (approval number 201100173).

**Sample selection.** The UK Biobank is a large prospective cohort study of ~500 K individuals from the United Kingdom[40]. Individuals are aged 40–69 years and are assessed for a range of traits; including physical, socio-economic and cognitive factors, as well as medical history. Most individuals have genotype information for 807,411 or 825,927 markers from the UK BiLEVE or UK Biobank Axiom Arrays[41]. We identified 417,060 individuals from this cohort that had (1) British or Western European ancestry (see details below), (2) consistent self-reported and genetic sex, (3) one or more recorded phenotypes for 32 quantitative or ordered categorical traits (see details below), (4) self-reported as born in Great Britain with coordinates for place of birth, (5) born between 1937 and 1970, (6) aged between 40 and 70 at the time of assessment, and (7) had imputed genotypes[41].

**Determining ancestry**. Genotype markers for the UK Biobank sample were quality checked and imputed to the Haplotype Reference Consortium (HRC) and UK10K reference panels by Bycroft et al.[41]. From these data, we hard-called 1,326,701 bi-allelic HapMap3 SNPs imputed from the HRC reference panel with imputation quality ≥0.3, minor allele count (MAC) > 5 and missingness <0.05 using PLINK (v200aLM)[42]. We then used a multi-step, iterative process to quality check and identify individuals of British or Western European ancestry using the 2,504 participants in the 1,000 Genomes Project[43] with known ancestries as a reference. First, we identified 1,029,456 variants in common with the 1,000 Genomes Project and with minor allele frequency (MAF) > 0.01 in both datasets. Then the UK Biobank participants were projected onto the first two principal components (PC) from the 1000 Genomes Project reference panel using GCTA (v1.9)[5]. We classified Europeans as those with >0.9 probability as belong to the European supercluster based on the projection. Variants were then filtered for Hardy–Weinberg equilibrium in this tentative European subset (pHWE > 10⁻⁵), and the projection and assignments to the European cluster repeated. The resulting classification assigned 456,426 individuals to European ancestry. Next we repeated the above procedure within the European subset of the 1000 Genomes panel to obtain individuals with >0.9 probability of clustering with the GBR (British in England and Scotland) and CEU (Northern and Western European ancestry) ancestry individuals. Using this more stringent criterion, we obtained 449,298 individuals of likely GBR and CEU ancestry.

**Estimation of genomic relationships**. The genomic relationship matrix (GRM) was constructed using GCTA (v1.9)[5] for European ancestry individuals from a set of 1,123,347 HapMap3 SNPs (MAF > 0.01, HWE > 10⁻⁶ and missingness <0.05) originating from the Haplotype Reference Consortium (HRC) imputation panel. From this matrix, we used the --rel-cut-off option in GCTA to identify a subset of 348,502 individuals with a maximum genomic relationship ($\pi$) of 0.05 and a further set of 133,387 individuals with a maximum $\pi$ of 0.02.

**Genetic stratification**. Principal components were calculated with 137,102 genotyped SNPs using flashPCA (v2.0)[44] in unrelated individuals with European ancestry ($\pi < 0.05$). Genotyped SNPs were those previously used by the UK Biobank to calculate principal components[41], with some additional quality control filters (missingness < 0.05; pHWE 10⁻⁵; MAF 0.01). The resulting SNP loading were used to project all individuals onto the PC space. PC projections were conducted specifically in the European ancestry subset to capture genetic stratification related to the UK Biobank data.

**UK Biobank phenotypes**. Thirty-two quantitative and ordered categorical traits from the UK Biobank were selected from those available that had a high proportion of individuals with records. Phenotypes include anthropometric (standing height, waist:hip ratio, body mass index, body fat percentage, sitting:standing height ratio) and blood traits (white blood cell count, red blood cell count, platelet count, eosinophil count); educational attainment, household income, spirometry (forced vital capacity, forced expiration volume) and some sex-limited traits (relative age at voice break, age at menarche, age at first birth). A full listing of traits, UK Biobank field identifiers and the number of records analysed are given in Supplementary Table 1.

**Geographic stratification (birth contemporary groups)**. The Geographic Information Systems (GIS) shapefile containing the boundaries of local authority areas in UK were obtained from the InFuse website[45], which is part of the UK Data Servide Census Support. The R-packages sp (v1.4-4) and rgdal (v1.5-18) were used to merge the spatial data from local authority GIS shapefile[18,46,47] with the birth place coordinates of the UK Biobank participants (see Supplementary Table 2) in order to create 378 contemporary groups. Birth CG was fitted to phenotypes (see below) to assess and account for common environmental effects acting at the level of local authorities.

**Models fitted to the phenotypes**. Four models were fitted to each phenotype. Fixed effects included in all models as factors were sex (2 levels), genotyping batch (batch, 106 levels), year of birth (yob, 34 levels) and age at assessment (age, 31 levels). Models differed in the degree of geographic or genetic stratification in the model. A list of fixed effects and UK Biobank identifiers can be found in Supplementary Table 2. The models fitted were:

$$y = mu + sex + batch + age + yob \qquad (1)$$

$$y = mu + sex + batch + age + yob + CG \qquad (2)$$

$$y = mu + sex + batch + age + yob + PC1 + PC2 + \ldots + PC25 \qquad (3)$$

$$y = mu + sex + batch + age + yob + PC1 + PC2 + \ldots + PC25 + CG \qquad (4)$$

Observations more than 5 standard deviations from the mean were excluded from further analysis. Residuals were normalised to have mean zero and unit variance within each sex, and this is the definition of phenotype in this study.

**Phenotypic covariance**. The $8.6 \times 10^{10}$ elements of the genomic relationship matrix represent all the pairwise relationships ($\pi$) between the 417,060 individuals with data in our British or Western European ancestry subset of the UK Biobank sample. We excluded pairs known to be direct descendants (i.e. 6276 parent-offspring pairs identified by Bycroft et al.[41]) due to the different expected covariance under assortative mating[2]. Covariance bins were constructed based on the distribution of the observed relationships, as a trade-off between accuracy of estimates (tending to increase bin width) and resolution (tending to decrease bin width). For each bin, the average phenotypic covariance for all pairs in a bin was calculated as $\frac{1}{N_k} \sum_i^y y_j$, where $y_i$ and $y_j$ are the residual standardised phenotypes for individual $i$ and $j$, and $N_k$ is the number of pairs in bin $k$. Standard errors were calculated using a blocked Jackknife approach with 100 blocks of individuals.

**SNP-based heritability in unrelated pairs of individuals**. The SNP-based heritability, the proportion of variance captured by common SNPs ($\hat{h}_{SNP}^2$) estimated using unrelated individuals, was calculated two ways. We either used a weighted linear regression in R[18] (where weights were equal to $N_k$ pairs per bin) of the phenotypic covariance on the average genomic relationship per bin or individual-level cross-product HE regression in GCTA (v1.9)[7]. We used a maximum relationship threshold of 0.02 ($\pi < 0.02$).

**Heritability in relative pairs**. We estimated the heritability using relatives ($\pi > 0.05$) in two ways. First we used a weighted linear regression in R[18] (where weights were equal to $N_k$) of the phenotypic covariance per bin on the average genomic relationship. This weighted HE regression is similar to an analysis based on expected (pedigree) relationships, under the assumption that $\pi$ centres around the expected relationship between pairs. Second, we used the cross-product HE regression in GCTA (v1.9)[7] with two variance components following a similar approach to Zaitlen et al.[9] The first component captured the covariance due to close relative pairs by (i) extracting a genomic relationship matrix consisting of a subset of 197,173 individuals with one or more close relatives in the dataset and (ii) modifying the matrix by setting all the small relationships ($\pi < 0.05$) to zero using the --make-bK option in GCTA (v1.9)[7]. This relationship matrix could be thought of as similar to a (realised) pedigree matrix. The second component fitted the average covariance for all pairs (equivalent to an intercept) by creating a second matrix identical to the modified matrix described above, and then setting any non-zero off-diagonal elements to 1.

**Accounting for assortative mating**. Under positive assortative mating, the additive genetic variance increases compared to a random mating population until a steady state is reached. For known relatives, the increase in genetic variance is a function of $rh_{EQ}^2$ and $d$, where $r$ is the phenotypic correlation between mates, $d$ is the number of meiosis separating a pair of relatives and $h_{EQ}^2$ is the equilibrium heritabilty[2]. Yengo and Visscher[48] gave a general approximation of the relationship between the phenotypic covariance of relatives $i$ and $j$ under assortative mating, assuming that the only contribution to this covariance is additive genetic variance and that the phenotypic variance in the equilibrium population is unity, then

$$\mathrm{cov}\left(y_i, y_j \mid d, r, h_{EQ}^2\right) \approx (0.5)^d h_{EQ}^2 \left(1 + rh_{EQ}^2\right)^d \qquad (5)$$

If we replace $(0.5)^d$ by $\pi_k$, for pairs of relatives in the genomic relationship bin $k$, then $d_k = \log(\pi_k)/\log(0.5)$, and $\mathrm{cov}(y_i, y_j \mid \pi_k) \approx \pi_k h_{EQ}^2 (1 + rh_{EQ}^2)^{d_k}$. Defining $y_k = \frac{cov(y_i, y_j \mid \pi_k)}{\pi_k}$, we then have a linear model,

$$\log(y_k) = \alpha + \beta d_k + e = \alpha + \beta \frac{\log(\pi_k)}{\log(0.5)} + e, \qquad (6)$$

with $\alpha = \log(h_{EQ}^2)$ and $\beta = \log(1 + rh_{EQ}^2)$. Setting the phenotypic variance equal to 1, we assessed the evidence for assortative mating for heritable traits ($\hat{h}_{SNP}^2 > 0.10$) with more than 400 K records per trait. We use data for 7 relative pair bins with mean $\pi > 0.05$ and solve the above equation as $h_{EQ}^2 = e^a$ and $r = (e^b - 1)/e^a$, where $a$ and $b$ are estimates of $\alpha$ and $\beta$. Standard errors were estimated using a block Jackknife approach with 100 blocks of individuals.

**Full-sib IBD regression analysis**. We used full-sib IBD regression[11] to estimate the genetic variance in the absence of environmental covariance. Simulation verified that heritability obtained from full-sib IBD regression is $\sigma_{a(RM)}^2 / \sigma_{P(EQ)}^2$, where $\sigma_{a(RM)}^2$ is the genetic variance in a random mating population and $\sigma_{P(EQ)}^2$ is the phenotypic variance under equilibrium conditions (Supplementary Note 3). We identified 20,342 pairs consisting of 36,920 individuals in our British and Western European dataset that were classified as full-sibs by Bycroft et al.[41]. Pairs were from 17,880 inferred families, with up to 6 members per family. Then, following Hemani et al.[22], the proportion of alleles IBD was estimated for each pair using Merlin (v1.1.2)[49]. Briefly, estimating the proportion of alleles IBD involved pruning the HM3 SNPs into an informative subset of 25,355 autosomal SNPs (MAF > 0.10, $r^2 < 0.05$ in 5 Mb windows and sliding in 2.5 Mb chunks across the genome), estimating IBD probabilities for each variant and combining variants into genome-wide

estimates using genetic (recombination) distance. The genetic map was obtained from the 1000 genomes phase 1 website [URL: ftp://ftp.1000genomes.ebi.ac.uk/vol1/ftp/technical/working/20130507_omni_recombination_rates/]. SNPs with duplicate positions were excluded from the analysis and we used allele frequencies calculated from the 348,502 unrelated Europeans subset as input into Merlin. IBD estimates from Merlin showed a strong correlation (0.98) with the KING kinship estimates provided by the UK Biobank (Supplementary Figure 8), with the distribution of IBD estimates in agreement with expectations[11] (mean 0.50, SD 0.037).

Following Hemani et al.[22], the full-sib IBD regression was conducted using a mixed linear model with variance components estimated via REML in GCTA (v1.9)[5]. REML was chosen over a simple full-sib covariance regression as all data and relationships within families are included in the model. We fitted the following model to the data:

$$\mathbf{y}^* = \mathbf{1}\mu + \mathbf{a} + \mathbf{c} + \mathbf{e} \qquad (7)$$

where $\mathbf{y}^*$ is a vector of residuals from (1) above (i.e. corrected for sex, yob, age and genotyping batch), $\mathbf{1}$ is a vector of ones, $\mu$ is the mean, $\mathbf{a}$ is a vector of additive genetic effects, distributed $\mathbf{a} \sim N(0, \mathbf{A}\sigma^2_a)$, $\mathbf{c}$ is a vector of family effects, distributed $\mathbf{c} \sim N(0, \mathbf{C}\sigma^2_c)$ and $\mathbf{e}$ is a vector of residual errors, distributed $\mathbf{e} \sim N(0, \mathbf{I}\sigma^2_e)$. Covariance matrix $\mathbf{A}$ is block diagonal, with 1's on the diagonal and genome-wide proportion of IBD on off-diagonals for full-sib pairs. $\mathbf{C}$ is also a block diagonal matrix with 1's on the diagonal and 1's on off-diagonal elements for full-sib pairs. $\mathbf{I}$ is an identity matrix. The equivalence between the pair-based full-sib regression and Eq. (7) is detailed in Supplementary Note 2.

The meta-analysis of full-sib regression results was conducted using the inverse-weighted approach described by Hemani et al.[22]. Full-sib correlations ($\hat{r}_{FS}$) were calculated as $\hat{c}^2_{FS} + \frac{1}{2}\hat{h}^2_{FS}$ for all studies, with standard errors approximated following Fisher[50] as $\sqrt{(1 - \hat{r}^2_{FS})/N}$, where $N$ is the number of pairs contributing to the estimate.

## Data availability

UKBiobank: Raw data from this study is available from the UK Biobank. Data access policies (http://www.ukbiobank.ac.uk/register-apply/) and a description of the genetic data (http://www.ukbiobank.ac.uk/scientists-3/genetic-data/) are available from the UK Biobank website. The UK Biobank data is available to all bona fide researchers. 1000 Genomes data: URL: ftp://ftp.1000genomes.ebi.ac.uk/vol1/ftp/phase3/; 1000 Genomes genetic map: URL: ftp://ftp.1000genomes.ebi.ac.uk/vol1/ftp/technical/working/20130507_omni_recombination_rates/; Results: Data and scripts to reproduce all figures and tables are provided in the Supplementary Data.zip file. Source data are provided with this paper.

## Code availability

Most software programs used in this study are publicly available: GCTA (https://cnsgenomics.com/software/gcta/#Overview), flashPCA (https://github.com/gabraham/flashpca), PLINK (https://www.cog-genomics.org/plink/2.0/ and https://www.cog-genomics.org/plink/1.9/), R (https://www.r-project.org/), Rstudio (https://www.rstudio.com/) and Merlin (https://csg.sph.umich.edu/abecasis/Merlin/download/). The Fortran source code to calculate the average phenotypic correlation in genomic relationship bins is provided in the Source data file.

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

## Acknowledgements

This research has been conducted using the UK Biobank Resource under project 12514. This research was supported by the Australian National Health and Medical Research Council (NHMRC: 1103418, 1107258, 1127440, 1113400), the Australian Research Council (ARC: DP160102400, FT180100186, FL180100072) and the National Institutes of Health (NIH: R01AG042568 and R01MH100141). We thank Dr Alexander Young for supplying additional results from his paper[12].

## Author contributions

K.E.K. and P.V.M. conceived and designed the analyses. K.E.K. performed the analyses. L.Y., Z.Z., A.A., M.C.K, M.E.G., N.W. and J.Y. contributed to data quality control, local authority birth contemporary groups, simulations, construction of the genomic relationship matrix and guidance for analysis. K.E.K. and P.V.M. wrote the manuscript, with participation of all authors.

## Competing interests

The authors declare no competing interests.
