## [Peer Review File · Nature Communications]

Reviewer #3 (Remarks to the Author):

The authors have addressed my remaining concerns and substantially improved the presentation of the results. I hope that one day we will be able to settle these controversies not only for height, but for education related traits also.

Reviewer #4 (Remarks to the Author):

I was asked, primarily, to look at the comments made by Reviewer 2 and indicate whether they had been appropriately addressed in the revised manuscript.

The two principal concerns of Reviewer 2 are: 1) the distinction between analysis that is IBS (most of the presented results) and IBD and its implications, and 2) the possibly misleading representation of results regarding educational attainment (EA) – a potentially politically sensitive issue – due, in part, to the large standard errors in estimation, but also failure to account for genetic nurturing (which the authors include under ‘associative effects’). In response to 1) the authors have striven to clarify the IBS/IBD distinction in several places in the text and have changed the h^2 PED notation to $h^2_{pi>0.05}$ to avoid confusion. In response to 2) the authors have limited most of their discussion to the analyses involving height, rather than EA and explicitly acknowledge that the EA estimates have large standard errors and that assortative mating and other associative effects may well have an impact on the EA estimation. The revisions seem to address these two principal concerns appropriately (even if more could have been done).

Most of the specific concerns of Reviewer 2 are addressed by the authors reasonably (exceptions follow).

The authors did not feel that they could reasonably exclude portions of the manuscript (possibly for a separate publication) due to their conviction that it would impede the understanding of the flow of the analysis. They have attempted to put details that would otherwise distract from the main results into supplementary material. While I doubt that this would satisfy Reviewer 2, it does not seem to be his principal concern and though I concur with the reviewer that the paper can be a bit hard to follow, I do not think it should preclude publication.

The authors also felt that additional IBD-based analyses in their close relative pairs (suggested by the reviewer) would both not be particularly easy to perform and would only add to the manuscript’s complexity. Given that the authors choose not go through this additional work, it seems that they could have indicated in the discussion that such an analysis could provide insight in future investigations.

I cannot speak for Reviewer 2, but for myself, I consider the paper to be publishable.

We thank the reviewers for their comments and address their comments below. Key changes to the manuscript in response to the reviewers' comments are highlighted (in yellow) in the manuscript.

REVIEWER #3:

The authors have addressed my remaining concerns and substantially improved the presentation of the results. I hope that one day we will be able to settle these controversies not only for height, but for education related traits also.

We thank the reviewer for their time and thoughtful comments.

Reviewer #4:

I was asked, primarily, to look at the comments made by Reviewer 2 and indicate whether they had been appropriately addressed in the revised manuscript.

The two principal concerns of Reviewer 2 are: 1) the distinction between analysis that is IBS (most of the presented results) and IBD and its implications, and 2) the possibly misleading representation of results regarding educational attainment (EA) – a potentially politically sensitive issue – due, in part, to the large standard errors in estimation, but also failure to account for genetic nurturing (which the authors include under 'associative effects'). In response to 1) the authors have striven to clarify the IBS/IBD distinction in several places in the text and have changed the h^2 PED notation to $h^2_{pi>0.05}$ to avoid confusion. In response to 2) the authors have limited most of their discussion to the analyses involving height, rather than EA and explicitly acknowledge that the EA estimates have large standard errors and that assortative mating and other associative effects may well have an impact on the EA estimation. The revisions seem to address these two principal concerns appropriately (even if more could have been done).

Thank you for acknowledging the changes we made in response to reviewer #2's comments.

Most of the specific concerns of Reviewer 2 are addressed by the authors reasonably (exceptions follow).

The authors did not feel that they could reasonably exclude portions of the manuscript (possibly for a separate publication) due to their conviction that it would impede the understanding of the flow of the analysis. They have attempted to put details that would otherwise distract from the main results into supplementary material. While I doubt that this would satisfy Reviewer 2, it does not seem to be his principal concern and though I concur with the reviewer that the paper can be a bit hard to follow, I do not think it should preclude

publication.

The authors also felt that additional IBD-based analyses in their close relative pairs (suggested by the reviewer) would both not be particularly easy to perform and would only add to the manuscript's complexity. Given that the authors choose not to go through this additional work, it seems that they could have indicated in the discussion that such an analysis could provide insight in future investigations.

Thank you. We have added to the discussion (paragraph 9) the following text:

'Our paper makes use of identical-by-state (IBS) relationships to estimate heritability in close relatives. Ideally, these inferences would be made based on the proportion of the genome identical-by-descent (IBD, where IBD alleles are IBS and inherited from a common ancestor³⁵) rather than IBS relationships³⁶. IBD relationships are ideal because (true) IBD sharing is independent of the genotyped markers. However, there are challenges associated with the accurate estimation of the proportion IBD in relatives without pedigree information. Hill and White³⁷, for example, highlight that the proportion IBD estimated from the detection of shared segments can underestimate the true value when small segments are missed. The estimation and use of IBD and IBS relationships in close and distant relatives warrants further scrutiny in the future.'

I cannot speak for Reviewer 2, but for myself, I consider the paper to be publishable.

Thank you for your time to review our manuscript.